# Short-Term Pharmacological Induction of Arterial Stiffness and Hypertension with Angiotensin II Does Not Affect Learning and Memory and Cerebral Amyloid Load in Two Murine Models of Alzheimer’s Disease

**DOI:** 10.3390/ijms23052738

**Published:** 2022-03-01

**Authors:** Jhana O. Hendrickx, Elke Calus, Peter Paul De Deyn, Debby Van Dam, Guido R. Y. De Meyer

**Affiliations:** 1Laboratory of Physiopharmacology, University of Antwerp, 2610 Antwerp, Belgium; jhana.hendrickx@uantwerpen.be; 2Laboratory of Neurochemistry and Behaviour, University of Antwerp, 2610 Antwerp, Belgium; elke.calus@uantwerpen.be (E.C.); peter.dedeyn@uantwerpen.be (P.P.D.D.); debby.vandam@uantwerpen.be (D.V.D.); 3Department of Neurology and Alzheimer Center, University of Groningen, 9713 GZ Groningen, The Netherlands; 4University Medical Center Groningen, 9700 RB Groningen, The Netherlands

**Keywords:** arterial stiffness, hypertension, angiotensin II, Alzheimer’s disease, cognition, amyloid

## Abstract

Given the unprecedented rise in the world’s population, the prevalence of prominent age-related disorders, like cardiovascular disease and dementia, will further increase. Recent experimental and epidemiological evidence suggests a mechanistic overlap between cardiovascular disease and dementia with a specific focus on the linkage between arterial stiffness, a strong independent predictor of cardiovascular disease, and/or hypertension with Alzheimer’s disease. In the present study, we investigated whether pharmacological induction of arterial stiffness and hypertension with angiotensin II (1 µg·kg^−1^·min^−1^ for 28 days via an osmotic minipump) impairs the progression of Alzheimer’s disease in two mouse models (hAPP23^+/−^ and hAPPswe/PSEN1dE9 mice). Our results show increased arterial stiffness in vivo and hypertension in addition to cardiac hypertrophy after angiotensin II treatment. However, visuospatial learning and memory and pathological cerebral amyloid load in both Alzheimer’s disease mouse models were not further impaired. It is likely that the 28-day treatment period with angiotensin II was too short to observe additional effects on cognition and cerebral pathology.

## 1. Introduction

The average age of the world’s population is steadily increasing. The number of people aged 65 and older is estimated to reach 1.4 billion in 2030 and 2.1 billion in 2050, up from 1 billion in 2020 [1]. This unprecedented rise in the world’s elderly population will increase the prevalence of age-related disorders, such as cardiovascular disease (CVD) and dementia. Globally, CVD was the number one cause of death in 2019 with an estimated death toll of 17.9 million people, of which 85% were due to heart attack and stroke [2]. Dementia, on the other hand, was the seventh leading cause of death in 2019 with an estimated death toll of 1.6 million people, with Alzheimer’s disease (AD) being the most prominent dementia syndrome [3] presenting a broad range of symptoms from motor dysfunctions to psychobehavioral manifestations [4]. The most common early manifestations of AD are loss of short-term memory, visuospatial dysfunction, impaired reasoning and language dysfunction, with cerebral β-amyloid accumulation and neurofibrillary tangles as the two major neuropathological hallmarks of the disease [5,6].

Historically, CVD and AD were considered separate entities based on clinical classification criteria. However, an increasing number of epidemiological studies report an independent convergence between the two diseases, suggesting a mechanistic overlap. Hypertension is the most prevalent form of CVD occurring in nearly one-third of adults and two-thirds of older adults worldwide [7,8] and is classified as a risk factor for AD [9]. More specifically, clinical research has associated hypertension with increased cerebral amyloid-β (Aβ) plaque burden [10,11,12], neurofibrillary tangle density [13,14,15] and cerebral atrophy [16]. Accordingly, recent research on the impact of hypertension on post-mortem AD neuropathology has shown that hypertension increases AD neuropathology indirectly through its effect on atherosclerosis of the circle of Willis. The authors presumed that this effect is due to persistent cerebral hypoperfusion leading to Aβ production and tau phosphorylation or diminished clearance of Aβ [7]. Moreover, we have previously demonstrated that altered stress hormone levels affect in vivo vascular function in the hAPP23^+/−^ overexpressing AD mouse model [17]. Therefore, it has been argued that mental stress is an important factor not only in CVD, e.g., hypertension but also in neurodegenerative diseases, such as AD [18,19,20,21,22].

In addition, previous research demonstrated that blood pressure elevation appears decades before the onset of AD, followed by a gradual reduction in blood pressure years before the onset of AD [23,24]. This phenomenon makes studying the link between hypertension and AD challenging and less accurate. In this perspective, class-specific and dose-dependent antihypertensive therapies have been shown to reduce AD pathogenesis [25]. However, a global clinical trial, which included 18,017 hypertensive patients, concluded that only 32% of patients treated with antihypertensive drugs achieved systolic blood pressure (SBP) control [26]. Later, the REASON study [27] was able to explain this poor clinical outcome by finding a positive correlation between SBP and arterial stiffness (AS), a strong independent predictor of CVD.

There is evidence that increased pulse wave velocity (PWV), an in vivo marker of AS, is associated with more rapid cognitive decline [6,28] and neuroanatomical changes associated with AD [29,30,31]. Hughes et al., showed that AS, as measured by PWV, is associated with the amount of cerebral Aβ deposition and that AS is an independent predictor of progressive Aβ deposition [32]. In contrast to blood pressure measurements, PWV increases [33] in a sustained manner over several decades prior to the diagnosis of dementia [34], suggesting that AS is a driving factor linking hypertension, cognitive impairment and consequent neuroanatomical changes.

The aim of the present study was to investigate whether pharmacological induction of AS and hypertension with angiotensin II (AngII) is able to alter visuospatial learning and memory and cerebral amyloidosis in two different AD mouse models. Moreover, we aimed to investigate in depth AS, both in vivo and ex vivo, in relation to the imposed AngII treatment in these mouse models. A schematic overview of the study design is shown in Figure 1. From a translational point of view, we study the exposure to AngII in adult rodents (6 months of age) to mimic early-onset hypertension, since early-onset rather than late-onset hypertension appears to be related to mild cognitive impairment [35].

## 2. Results

### 2.1. AngII Treatment Affects hAPP23^+/−^ and hAPPswe/PSEN1dE9 Mice Differently

Assessment of general characteristics of the mice showed significantly decreased body weights for hAPP23^+/−^ mice (*p* < 0.001) compared to their WT controls in contrast to hAPPswe/PSEN1dE9 animals, which had almost the same weight compared to their WT controls (Table 1). AngII treatment reduced body weight in hAPPswe/PSEN1dE9 animals and their WT controls (*p* = 0.005, Table 1). Interestingly, hAPPswe/PSEN1dE9 WT mice were significantly more prone to develop abdominal aortic aneurysms (AAA) compared to hAPPswe/PSEN1dE9 (*p* < 0.0001) whose incidence was similar in hAPP23^+/−^ mice and their WT controls (Table 1).

### 2.2. AngII Treatment Induces Hypertrophic Cardiomyopathy

Echocardiography revealed signs of cardiac hypertrophy in hAPP23^+/−^ (Table 2) and hAPPswe/PSEN1dE9 AngII-treated mice (Table 3) although the effect of AngII treatment was more pronounced in AngII-treated hAPPswe/PSEN1dE9 animals. Signs of cardiac hypertrophy included increased heart weights, left ventricular masses, inner-ventricular septum thicknesses and left ventricular posterior wall thicknesses. In addition, evidence of decreased diastolic heart function was obtained for the hAPP23^+/−^ murine model by means of increased E/E’ ratios (*p* = 0.008), IVRT and deceleration times.

### 2.3. AngII Treatment Induces Hypertension

Peripheral blood pressure measurements showed hypertension by significant increases in systolic blood pressure (Figure 2A,D; *p* < 0.001 and *p* < 0.001), diastolic blood pressure (Figure 2B,E; *p* < 0.001 and *p* < 0.001) and pulse pressures (Figure 2C,F; *p* = 0.02 and *p* = 0.02) upon AngII treatment in both AD murine models. In contrast to the hAPPswe/PSEN1dE9 animals, hAPP23^+/−^ mice showed genotype-dependent increments in systolic blood pressure (*p* = 0.004), diastolic blood pressure (*p* = 0.01) and pulse pressure (*p* = 0.04). Furthermore, sham- and AngII-treated hAPPswe/PSEN1dE9 animals presented with decreased pulse pressures compared to WT controls (Figure 2F).

### 2.4. AngII Treatment Induces In Vivo Arterial Stiffness

In vivo assessment of AS upon AngII treatment showed significantly elevated aPWV in both AD murine models and their respective WT controls (Figure 3A,C; *p* < 0.001 and *p* < 0.001). However, AngII treatment of hAPPswe/PSEN1dE9 and their WT control animals resulted in a significantly increased incidence of AAAs, with the AAA incidence of AngII-treated WT control animals doubling compared to that of AngII-treated hAPPswe/PSEN1dE9 mice (Table 1), causing a significant genotypic effect (Figure 3C, *p* = 0.007). Ex vivo assessment of AS revealed elevated genotype-dependent Ep-values in hAPP23^+/−^ mice (Figure 3B; *p* = 0.01) but not in hAPPswe/PSEN1dE9 AD (Figure 3D). A genotype effect was observed in hAPPswe/PSEN1dE9 mice, as AngII-treated WT control animals showed significantly increased Ep-values compared to sham-treated WT control animals (Figure 3D, *p* < 0.0001). This observation parallels the high AAA incidence observed in these animals after AngII treatment (Table 1).

### 2.5. AngII Treatment Does Not Lead to Impaired MWM Acquisition Trial Performance

AngII treatment did not result in impaired MWM acquisition trial performance in terms of total path length and escape latency for both hAPP23^+/−^ (Figure 4A–C) and hAPPswe/PSEN1dE9 (Figure 4D–F) animals and their respective WT controls. Moreover, AngII treatment caused a slower swimming speed in hAPP23^+/−^ and their WT control mice (Figure 4C) in contrast to hAPPswe/PSEN1dE9 mice, which showed a faster swimming speed compared to WT controls after treatment (Figure 4F; *p* = 0.001).

### 2.6. AngII Treatment Does Not Result in Impaired Probe Trial Performances

Both AngII-treated and sham-treated hAPP23^+/−^ performed less during the MWM probe trial compared to WT controls (Figure 5A; *p* = 0.02) where the AngII treatment did not result in a worse performance. On the other hand, hAPPswe/PSEN1dE9 presented with a deteriorated performance compared to WT controls where no clear effect of the AngII treatment could be observed for the hAPPswe/PSEN1dE9 murine model and respective WT controls (Figure 5B). However, a trend toward increased deterioration of visuospatial learning and memory could be interpreted for the AngII-treated WT animals (Figure 5D; *p*_Dirichlet_= 0.002). Dirichlet analysis of the probe trial performances revealed a more randomized presence of AngII-treated hAPPswe/PSEN1dE9 animals in the MWM (Figure 5D; *p*_Dirichlet_ = 0.00014). The AngII treatment of hAPP24^+/−^ animals, on the other hand, seemed to have a lesser impact (Figure 5B).

### 2.7. AngII Treatment of AD Mice Does Not Result in Increased Amyloid Load

Cerebral amyloidosis was assessed via histological analysis of the hippocampus and cortex of hAPP223^+/−^ and hAPPswe/PSEN1dE9 mice. Overall, AngII treatment did not result in a significantly increased amyloid load in the hippocampus of both hAPP223^+/−^ and hAPPswe/PSEN1dE9 mice although an increased and decreased trend could be observed for hAPP223^+/−^ and hAPPswe/PSEN1dE9 mice, respectively (Figure 6A). However, these trends are attributed to the large distribution of data points. Similar results were obtained for the cortex, where no significantly increased amyloid burden could be observed in both AD murine models (Figure 6B). In addition, the cortical amyloid burden was higher in hAPP223^+/−^ brains compared to hAPPswe/PSEN1dE9 brains (Figure 5A,B).

## 3. Discussion

In the present study, we aimed to investigate the mechanistic convergence between pharmacologically induced AS and AD in two well-validated murine models of AD, i.e., hAPP23^+/−^ and hAPPswe/PSEN1dE9, on a C57BL/6 background. As expected, treatment of animals with AngII resulted in hypertension, which was more pronounced in hAPP23^+/−^ animals than in hAPPswe/PSEN1dE9, compared to their respective controls. Similar conclusions could be drawn from the in vivo and ex vivo AS assessments where increased PWV and Ep measurements were obtained for the AD murine models, although this trend was less clear for hAPPswe/PSEN1dE9. A clear downstream effect of the prolonged AngII treatment was the development of hypertrophic cardiomyopathy, which was present in both AD murine models. In all cardiovascular assessments, it should be noted that remarkably higher measurements were obtained for the C57BL/6 control animals of the hAPPswe/PSEN1dE9 AD murine model. This outcome might be explained by the significantly increased AAA incidence in these animals, which is known to result in extreme hypertension and AS [37,38]. The reason why this C57BL/6 strain was more prone to developing AAA is unclear. It could be argued that the transportation of the animals prior to the test battery led to increased stress. High levels of cortisol, as seen in patients with Cushing’s syndrome, are known to be a risk factor for aortic aneurysms [39]. However, the above statement was ruled out following measurements of circulating serum corticosterone levels (data not shown) that revealed no significant increases in these mice compared to the other C57BL/6 strain and generally between control and AD mouse models.

Although in the present study short-term angiotensin II treatment induced hypertension and increased aPWV measurements, it had no additional effect on learning and memory and amyloid load in two AD mouse models. This outcome might be justified by the rather short 28-day AngII treatment period. Additionally, it should be noted that a rather short MWM protocol was used in this experiment and that animals were already tested after the two weeks of treatment, which might have been too early to notice significant behavioral changes. However, a similar neurobehavioral outcome was drawn in the past by Wiesmann et al., who performed a similar protocol on hAPPswe/PSEN1dE9 animals, but with only halve the AngII dosage and on 10-month-old animals [40]. Moreover, the authors were unable to find exacerbated amyloid pathology because of the AngII treatment, which is consistent with our assessment of cerebral amyloidosis. In contrast to this experiment, Wiesmann et al., used specific brain-imaging techniques, such as MRI and resting-state fMRI protocols, diffusion tensor imaging and a flow-sensitive alternating inversion recovery MRI protocol for cerebral blood flow measurements. The authors were able to detect decreased AD-like neuropathological changes like functional connectivity in both AngII-treated control and AD animals, whereby only AngII-treated AD animals showed an impaired cerebral blood flow [40]. Wiesmann et al., further illustrated this finding by showing that chronic AngII treatment only led to decreased hippocampal cerebral perfusion in AD animals [41]. As a result of their findings, the authors hypothesize that elevated AngII levels act solely as a driving force for the development of symptomatic AD disease. Although we obtained similar results in terms of MWM, our animals were half the age compared to those investigated by Wiesmann et al. A recent study of the relationship between the timing of hypertension and cognitive performance in middle age found that hypertension is more likely to affect cognitive function than structural brain changes, and at an early rather than late age [35]. Taking into account the neuropathological findings of Wiesmann et al., more specified neuropathological and brain imaging studies are required in the future to clarify neuropathological changes due to AngII-induced early-onset hypertension in our translational AD mouse model.

Recent research on the influence of AngII converting enzyme inhibition on AD symptoms in different experimental AD models showed delayed AD-related hippocampal neurodegeneration, but also actively promoted hippocampal neuro-regeneration [42]. Another treatment approach involves blocking AngII type 1, type 2 and type 4 receptors, which has been shown to be effective in preventing and restoring cerebrovascular, neuropathological, and cognitive impairment, not only in AD murine models [43,44,45] but also in AD patients [16]. As mentioned earlier, the global I-SEARCH study, in which 18,017 hypertensive patients participated, showed that only one-third of patients treated with antihypertensive drugs gained SBP control [26], reinforcing the idea that for patients at risk of AD, treatments for AS rather than hypertension are needed. Furthermore, less than 50% of hypertensive patients achieve adequate blood pressure control despite multiple drug treatments, and up to 0.5% of patients are refractory to drug treatment. Recently, deep brain stimulation of cerebral target regions was found effective in the management of refractory hypertension [46]. On the other hand, (non)invasive brain stimulation was also found effective in the treatment of neurodegenerative conditions, such as AD [47,48,49]. Altogether, (non)invasive brain stimulation may offer an option for the treatment of refractory hypertension, and consequently, AS [50], as well as AD [47], while simultaneously treating the contribution of underlying mental stress.

Limitations of this study include the fact that only male mice were evaluated, as recent epidemiological estimates indicate that women over 65 years of age have a 1 in 6 chance of developing AD compared to 1 in 11 men [51], which is largely but not entirely due to the higher life expectancy in women [3]. Secondly, future experiments should include animals from the same animal facility, since the AngII in the current experiment affected the AD murine models and respective WT controls differently, making exact comparisons more difficult to interpret. In addition, future research should consider an experimental set-up with a longer AngII treatment duration in order to better study the effects of AS and hypertension on AD pathophysiology. Moreover, future experiments should be considered that investigate the effect of (non)invasive brain stimulation on reversing pharmacologically induced hypertension/AS with AngII in AD mouse models.

In conclusion, our results show a rather limited effect of AngII treatment on visuospatial learning and memory and cerebral amyloidosis in two well-known AD murine models. It is likely that the 28-day treatment period with angiotensin II was too short to observe additional effects on cognition and cerebral pathology.

## 4. Materials and Methods

### 4.1. Experimental Animals and Osmotic Minipump Implantation

In this study, the hAPP23^+/−^ murine (sham-treated: *n* = 10; AngII-treated *n* = 13) model, together with the respective control C57BL/6 littermates (sham-treated: *n* = 15; AngII-treated: *n* = 12) were bred and housed in the laboratory animal facility of the University of Antwerp. Humanized APPswe/PSEN1dE9 mice (sham-treated: *n* = 13; AngII-treated *n* = 10) with their respective C57BL/6 littermates (sham-treated: *n* = 14; AngII-treated *n* = 12) were obtained from the Jackson Laboratory (Maine, Sacramento, CA, USA). Genotypes of all animals were confirmed by polymerase chain reaction (PCR) on DNA of 4-week-old mice (Primer mutant forward: AAT TCG CCA ATG ACA AGA CG, primer wild-type forward: AGG GGA ACA AGC CCA GTA GT, primer reverse common: CTT GTC CCC TAG GCA CCT CT). All mice were socially housed in standard mouse cages with a maximum of eight animals per cage under conventional laboratory conditions with a constant room temperature (22 ± 2 °C), humidity (55 ± 5%) and an artificial day/night cycle of 12 h/12 h (lights on at 8 a.m.). Food and water were provided ad libitum. All animal experiments were approved by the Animal Ethics Committee of the University of Antwerp (ECD approval n° 2020/06) and were conducted in accordance with the EU Directive 2010/63/EU and in accordance with the Animal Research: Reporting of In vivo Experiments (ARRIVE) guidelines [52].

Per the experimental subgroup, the selection of animals was randomized. There was no increased mortality in this study. The hAPP23^+/−^, APPswe/PSEN1dE9 and respective control littermates were split into two main groups: (1) induced hypertension using AngII-infusion delivered by subcutaneously implanted osmotic minipumps and (2) controls with PBS infusion. At the age of 5 months, animals assigned to the AngII treatment group received AngII (1 µg·kg^−1^·min^−1^, Sigma-Aldrich, Burlington, MO, USA) for a period of 28 days via the implantation of osmotic minipumps (ALZET, Cupertino, CA, USA, model 1004). Control animals were sham-operated to receive minipumps with sterile PBS (Sigma-Aldrich, Burlington, MO, USA). Osmotic minipumps were implanted subcutaneously under isoflurane anesthesia (induction with 1.5% in O_2_, 1 L/min and maintenance with 3.5% in O_2_, 1 L/min (Forene, Abbvie, Lake Bluff, IL, USA)) throughout the complete procedure via a nose cone placed over the snout of the animal. Mice were placed in a prone position on a preheated platform (37 °C) with embedded ECG leads (VisualSonics, Toronto, ON, Canada). Heart rhythm, respiratory rate and body temperature were constantly monitored. A mid-scapular incision, at the backside of the animal, was made and a subcutaneous pouch was created with blunt dissection. After implantation of the osmotic minipump, the mid-scapular incision was disinfected and sealed with wound clips and 5.0 monofilament non-absorbable sutures (Ethilon*, Ethicon, Somerville, NJ, USA).

Two weeks post-implantation of the osmotic minipumps, the visuospatial learning and memory of animals was assessed via a Morris water maze (MWM) test. The week after, blood pressure measurements and an echocardiographic analysis were performed. In the fourth week post-operation, mice were humanely killed by perforation of the diaphragm while under deep anesthesia (sodium pentobarbital (Sanofi, Paris, France), 250 mg/kg, i.p. [53]) for further ex vivo assessment of AS and tissue collection.

A schematic overview of the experimental set-up can be consulted in Figure 1.

### 4.2. Visuospatial Learning and Memory

Spatial learning and memory functions were evaluated at the age of 6 months in the hAPP23^+/−^ mice (sham-treated: *n* = 10; AngII-treated *n* = 13) together with the respective control C57BL/6 littermates (sham-treated: *n* = 14; AngII-treated: *n* = 12) and APPswe/PSEN1dE9 mice (sham-treated: *n* = 13; AngII-treated *n* = 11) with their respective C57BL/6 littermates (sham-treated: *n* = 12; AngII-treated *n* = 7) by means of the MWM test [54,55]. The MWM consisted of a circular pool (diameter: 150 cm, height: 30 cm) filled with opacified water using non-toxic white paint and was kept at 25 °C. Invariable visual cues were placed around the pool. The MWM consisted of an acquisition phase and a probe trial. The acquisition phase was performed over a period of 4 days and consisted of 2 daily trial blocks (1 at 10:30 a.m. and 1 at 03:00 p.m.) of 4 trials with a 15 min inter-trial interval. During the acquisition phase, a round acrylic glass platform (diameter 15 cm) was placed 1 cm below the water surface on a fixed position in the center of one of the pool’s quadrants. Mice were placed in the water facing the wall and were recorded while trying to find the hidden platform for a maximum duration of 120 s. If the mouse was not able to reach the platform within 120 s, it was guided to the platform, where they had to stay for 15 s before being returned to their home cage. The starting positions varied in a semi-random order. The probe trial followed 4 days after the final acquisition trial. For this trial, the platform was removed, mice were placed in the MWM at a fixed position, and swimming trajectories were recorded for a period of 100 s. During both acquisition and probe trials, the animals’ trajectories were recorded using a computerized video-tracking system (Ethovision, Noldus, Wageningen, The Netherlands), with path length, escape latency, and swimming speed recorded. Spatial accuracy is expressed as the percentage of time spent in each quadrant of the MWM, i.e., the specific location of the platform during the acquisition phase. The experimenters were blind to the genetic or treatment status of all mice.

### 4.3. Blood Pressure Measurements

Peripheral blood pressure was measured using the non-invasive CODA tail-cuff blood pressure system (KENT Scientific CO., Torrington, CT, USA), as previously described [56]. Mice were immobilized in a Plexiglas restrainer and an occlusion cuff and volume pressure cuff were placed around the tail of the mouse, respectively. Voltage output from both cuffs was recorded and analyzed by a PowerLab signal transduction unit associated chart software (ADInstruments, Colorado Springs, CO, USA). To minimize discomfort for the animals and to increase the reliability of measurements, the blood pressure of animals was measured 3 days prior to the effective measurement on day 4. For each mouse, a measurement consisted of 15 cycles (approximately 15 min per animal) and the reported values consisted of averaged values measured on day 4.

### 4.4. Echocardiography

Echocardiography was performed with a high-frequency, high-resolution digital imaging platform with linear array technology and color Doppler mode for in vivo micro-imaging (Vevo^®^ 2100 Imaging System, FUJIFILM VisualSonics Inc., Toronto, ON, Canada). To assess systolic and diastolic heart function in mice, a high-frequency transducer probe (VisualSonics MS500D, FUJIFILM VisualSonics, Inc., Toronto, ON, Canada) with a frequency range of 18–38 MHz) was used to provide appropriate resolution and depth of penetration needed. Transthoracic echocardiograms were performed on anesthetized mice (induction with 1.5% in O_2_, 1 L/min and maintenance with 3.5% in O_2_, 1 L/min, Forene, Abbvie, Lake Bluff, IL, USA). Mice were placed on a preheated platform in a supine position in order to maintain their body temperature at 36–38 °C, which was constantly monitored throughout the whole procedure via an anal thermometer probe. Isoflurane concentrations were titrated (1–2%) during imaging to maintain the heart rate at 500 ± 50 beats/minute. Systolic left ventricular dimensions were acquired via short-axis M-mode images. To calculate the percentage of fractional shortening (FS%) and ejection fraction (EF%), end-systolic and end-diastolic dimensions along with end-systolic and end-diastolic volumes and stroke volume were recorded via short-axis M-mode images. Diastolic cardiac function was determined using color and Pulsed-Wave (PW) Doppler recordings of the trans-tricuspid flow. Reported cardiac parameters consist of averaged measurements of three consecutive M-mode and/or PW Doppler images.

### 4.5. Non-Invasive Pulse Wave Velocity (PWV) Measurements of the Aortic Abdominal Aorta (aPWV)

A high-frequency, high-resolution digital imaging platform (Vevo^®^ 2100 Imaging System, FUJIFILM VisualSonics Inc., Toronto, ON, Canada) was used on anesthetized mice (induction with 1.5% in O_2_, 1l/min and maintenance with 3.5% in O_2_, 1 L/min, Forene, Abbvie, Lake Bluff, IL, USA). to assess pulse wave velocity measurements of the abdominal aorta (aPWV). Body temperature was maintained at 36–38 °C and mice were continuously monitored, and isoflurane concentrations were titrated (1–2%) during imaging to maintain heart rates at 500 ± 50 beats/minute (bpm). PWV measurements were performed with a 24-MHz transducer (VisualSonics MS400, FUJIFILM VisualSonics, Inc., Toronto, ON, Canada) using the method developed by Di Lascio et al., (2014) [57]. In short, a 24-MHz transducer was positioned on the abdomen of the animal. B-mode images of 700 frames-per-second of the abdominal aorta and carotid artery were obtained using the EKV imaging mode to measure the aortic diameter (D). A pulse wave doppler tracing was obtained to measure aortic flow velocity (V). Velocity was plotted against the natural logarithm of the diameter, and the slope of the linear part of the resulting ln(D)-V loop was used to calculate PWV values using Matlab v2014 (MathWorks).

### 4.6. Rodent Oscillatory Tension Set-Up for Arterial Compliance (ROTSAC)

At sacrifice, the thoracic aorta was carefully removed and cleared of adherent tissue. Starting approximately two millimeters distal to the aortic arch, the descending thoracic aorta was cut into four segments of two millimeters length for further vascular reactivity and stiffness analyses. These segments were immediately immersed in Krebs Ringer (KR) solution (37 °C, 95% O_2_/5% CO_2_, pH 7.4) containing (in mmol/L): NaCl (118 mmol/L), KCl (4.7 mmol/L), CaCl_2_ (2.5 mmol/L), KH_2_PO_4_ (1.2 mmol/L), MgSO_4_ (1.2 mmol/L), NaHCO_3_ (25 mmol/L), CaEDTA (0.025 mmol/L), and glucose (11.1 mmol/L). The KR solution was continuously aerated with a 95% O_2_/5% CO_2_ gas mixture to maintain the pH at 7.4 and was replaced periodically to prevent glucose depletion. Aortic segments were mounted between two parallel wire hooks in 10 mL organ baths filled with KR solution (37 °C, 95% O_2_/5% CO_2_, pH 7.4). Diameter and estimates of transmural pressure were derived as described previously [58]. In short, force and displacement of the upper hook were measured with a force-length transducer connected to a data acquisition system (PowerLab 8/30 and LabChart Pro, ADInstruments Inc., Colorado Springs, CO, USA). Force and displacement were acquired at 0.4 kHz. To estimate the transmural pressure that would exist in the equilibrated vessel segment with the given distension force and dimensions, the Laplace relationship was used. All measurements were performed over a pressure range with pressure clamps between diastolic 80 to systolic 120 mm Hg at 10 Hz chosen to allow calculation of the Peterson modulus (Ep):$$\mathrm{Ep}=D_{0}\times \frac{\Delta P}{\Delta D}$$
with ∆D as the difference between systolic and diastolic diameter, ∆P is the pressure difference of 40 mmHg and D_0_ is the diastolic diameter. This pulse pressure difference of 40 mmHg, applied at a frequency of 10 Hz, corresponds to a physiological heart rate of 600 beats per minute in mice, was kept constant throughout the experiment. Measurements took on average 5–10 min.

### 4.7. Histology

Upon collection, brain tissue was fixed for 24 h in 4% formalin solution (BDH Prolabo, Leuven, Belgium), and subsequently dehydrated in 60% isopropanol (BDH Prolabo, Leuven, Belgium), followed by paraffin embedding. Serial cross-sections of hippocampal/cortical brain tissue were prepared for histological analysis. Positively stained percentage area fractions for Aβ were calculated using the ImageJ software (23). In short, slides were transferred to 100% ethanol twice for 3 min and then once through 95%, 70%, and 50% for 3 min. Endogenous peroxidase activity was blocked by incubating the sections for 10 min in a 3% H_2_O_2_ solution in methanol. Slides were next rinsed twice in PBS for 5 min followed by an antigen retrieval using 88% formic acid for 20 min at room temperature. Hereafter, tissue sections were treated with 100 µL of 1:200 diluted anti-Aβ_17–24_ antibody (BioLegend Cat. No. SIG-39200, San Diego) at 4 °C overnight. The next day, the slides were washed twice in PBS for 5 min and 100 µL of biotinylated secondary antibody was added to the sections on the slides followed by incubation in a humidified chamber at room temperature for 30 min. Slides were washed twice in PBS for 5 min and 100 µL of Sav-HRP conjugates was added to the sections for 30 min in a humidified chamber at room temperature protected from light whereafter the slides were washed twice in PBS for 5 min. Freshly made DAB solution (100 µL), consisting of 0.05% DAB and 0.015% H_2_O_2_ in PBS, was next applied to the sections to reveal the color of the antibody, followed by a double washing step in PBS for 5 min and a hematoxylin counterstain for 1–2 min. Finally, the slides were washed in running tap water for more than 15 min and tissue sections were dehydrated through four changes of alcohol (95%, 95%, 100%, and 100%) for 5 min each whereafter the slides were mounted. Microscopic images were obtained with Universal Grap 6.1 software using an Olympus BX4 microscope and quantified with ImageJ software. Cerebral amyloidosis was analyzed on positively stained percentage area fractions on three hippocampal and five cortical (parietal and occipital cortex) microscopic images.

### 4.8. Statistical Analysis

Data are presented as mean ± SEM unless otherwise indicated. A factorial ANOVA was performed with the factor’s ‘Treatment’, ‘Genotype’, ‘Treatment × Genotype’. Survival rates were analyzed with a Log-Rank Mantel-Cox test. MWM probe trial statistics were calculated with a factorial ANOVA for the factor’s ‘Quadrant × Treatment’, ‘Quadrant × Genotype’ and ‘Quadrant × Treatment × Genotype’ and via Dirichlet distributions, as previously described [36]. Throughout the manuscript, differences were considered significant at *p* < 0.05. Applied statistical analyses are indicated in the figure legends and were performed using GraphPad Prism (version 9.1.2 for Windows, GraphPad Software, San Diego, CA, USA).

## Figures and Tables

**Figure 1 ijms-23-02738-f001:**
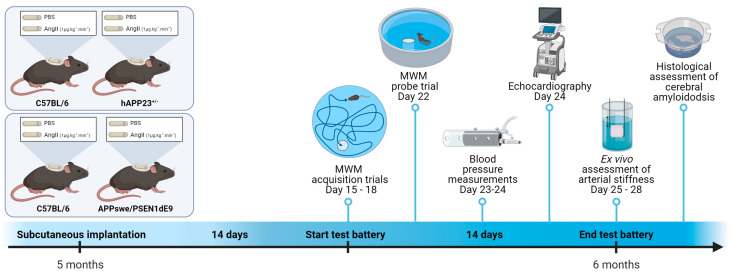
Schematic representation of the study design. Mice were (sham)-operated at the age of 5 months by subcutaneously implanting an osmotic minipump releasing AngII at a rate of 1 µg·kg^−1^·min^−1^. After approximately 14 days, the experimental test battery started with Morris water maze (MWM) trials on day 15 to day 18 followed by the MWM probe trial on day 22. On day 23 and day 24, blood pressure measurements with the CODA tail-cuff technique were performed, immediately followed by an echocardiographic analysis in the afternoon on day 24. From day 25 to day 28, the animals were humanely killed and ex vivo assessment of arterial stiffness and a histological assessment of cerebral amyloidosis were performed. This figure was created in BioRender (www.biorender.com, accessed on the 30 January 2022).

**Figure 2 ijms-23-02738-f002:**
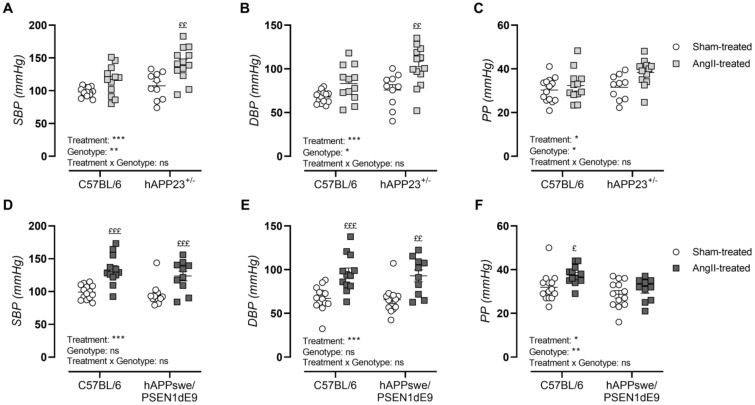
Peripheral blood pressure measurements. (**A**) Systolic blood pressure (SBP), (**B**) diastolic blood pressure (DBP) and (**C**) pulse pressures (PP) of sham- and AngII-treated C57BL/6 (sham: *n* = 14; AngII: *n* = 12) and hAPP23^+/−^ mice (sham: *n* = 13; AngII: *n* = 10). (**D**) SBP, (**E**) DBP and (**F**) PP of sham- and AngII-treated C57BL/6 (sham: *n* = 14; AngII: *n* = 12) and hAPPswe/PSEN1dE9 (sham: *n* = 13; AngII: *n* = 10). Factorial ANOVA for the factor’s ‘Treatment’, ‘Genotype’ and ‘Treatment × Genotype’, * *p* < 0.05, ** *p* < 0.01, *** *p* < 0.001 with a Sidak post hoc test, (^£^ *p* < 0.05, ^££^ *p* < 0.01, ^£££^ *p* < 0.001) compared to respective sham-treated animals. Data are presented as mean ± SEM.

**Figure 3 ijms-23-02738-f003:**
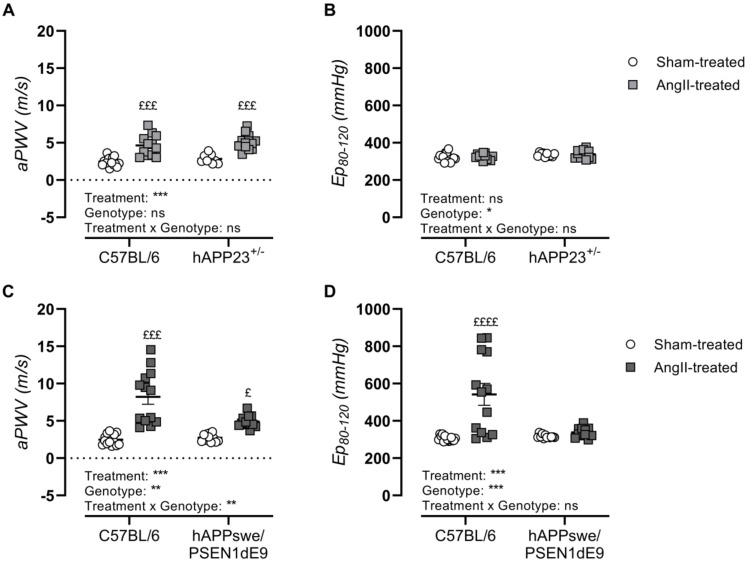
In vivo and ex vivo measurements of AS. (**A**) In vivo measurements of abdominal PWV (aPWV) and (**B**) ex vivo measurements of Ep over a pressure range of 80–120 mmHg of sham- and AngII-treated C57BL/6 (sham: *n* = 15; AngII: *n* = 12) and hAPP23^+/−^ animals (sham: *n* = 9; AngII: *n* = 13). (**C**) In vivo measurements of aPWV and (**D**) ex vivo measurements of Ep over a pressure range of 80–120 mmHg of sham- and AngII-treated C57BL/6 (sham: *n* = 14; AngII: *n* = 13) and hAPPswe/PSEN1dE9 animals (sham: *n* = 11; AngII: *n* = 12). Factorial ANOVA for the factor’s ‘Treatment’, ‘Genotype’ and ‘Treatment × Genotype’ (* *p* < 0.05, ** *p* < 0.01, *** *p* < 0.001) with a Sidak post hoc test (^£^ *p* < 0.05, ^£££^ *p* < 0.001, ^££££^ *p* < 0.0001) compared to respective sham-treated animals. Data are presented as mean ± SEM.

**Figure 4 ijms-23-02738-f004:**
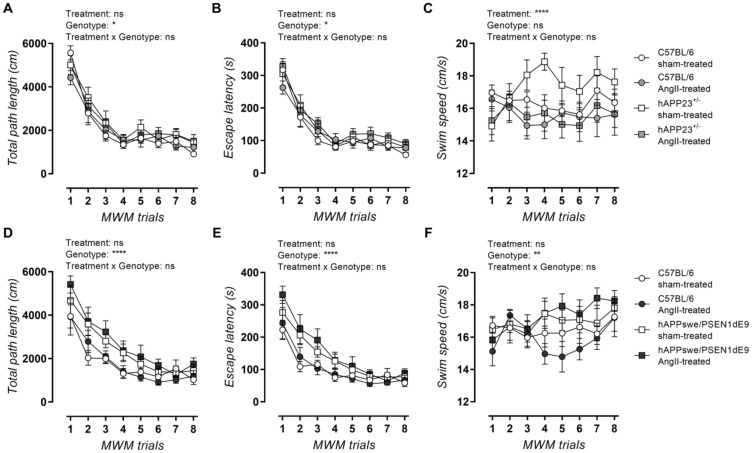
MWM acquisition trial results. (**A**) Total path length, (**B**) escape latencies and (**C**) swimming speed of sham- and AngII-treated C57BL/6 (sham: *n* = 14; AngII: *n* = 12) and hAPP23^+/−^ animals (sham: *n* = 10; AngII: *n* = 13). (**D**) Total path length, (**E**) escape latencies and (**F**) swimming speed of sham- and AngII-treated C57BL/6 (sham: *n* = 12; AngII: *n* = 7) and hAPPswe/PSEN1dE9 animals (sham: *n* = 11; AngII: *n* = 11). Factorial ANOVA for the factor’s ‘Treatment’, ‘Genotype’ and ‘Treatment × Genotype’ (* *p* < 0.05, ** *p* < 0.01, **** *p* < 0.0001). Data are presented as mean ± SEM.

**Figure 5 ijms-23-02738-f005:**
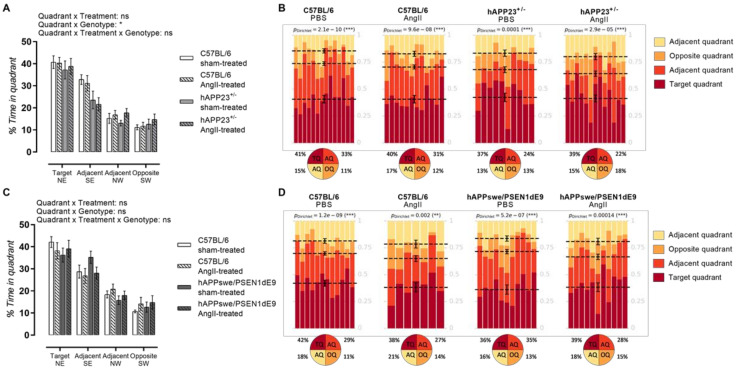
MWM probe trial results. (**A**) MWM probe trial of sham- and AngII-treated C57BL/6 (sham: *n* = 14; AngII: *n* = 12) and hAPP23^+/−^ animals (sham: *n* = 10; AngII: *n* = 13). (**B**) Statistical Dirichlet distributions of probe trial performances per tested group. (**C**) MWM probe trial of sham- and AngII-treated C57BL/6 (sham: *n* = 12; AngII: *n* = 7) and hAPPswe/PSEN1dE9 animals (sham: *n* = 11; AngII: *n* = 11. (**D**) Statistical Dirichlet distributions of probe trial performances per tested group. Factorial ANOVA for the factor’s ‘Quadrant × Treatment‘, ‘Quadrant × Genotype′ and ‘Quadrant × Treatment × Genotype’; (* *p* < 0.05, ** *p* < 0.01, *** *p* < 0.001. Dirichlet distributions were calculated as previously described [36]. (C57BL:6: sham-treated: *n* = 14; AngII-treated, *n* = 1. hAPP23^+/−^: sham-treated: *n* = 10; AngII-treated, *n* = 13|C57BL/6: sham-treated: *n* = 12; AngII-treated, *n* = 7. hAPPswe/PSEN1dE9: sham-treated: *n* = 11; AngII-treated: *n* = 11). Each column represents the probe trial performance of a single animal and each color represents a different quadrant. Mean values for the fraction of time spent in each quadrant are represented by a dotted line with respective error bars for SEM. Average percentages of time spent in each quadrant are represented in pie charts beneath the calculated heatmap of each group. Data are presented as mean ± SEM.

**Figure 6 ijms-23-02738-f006:**
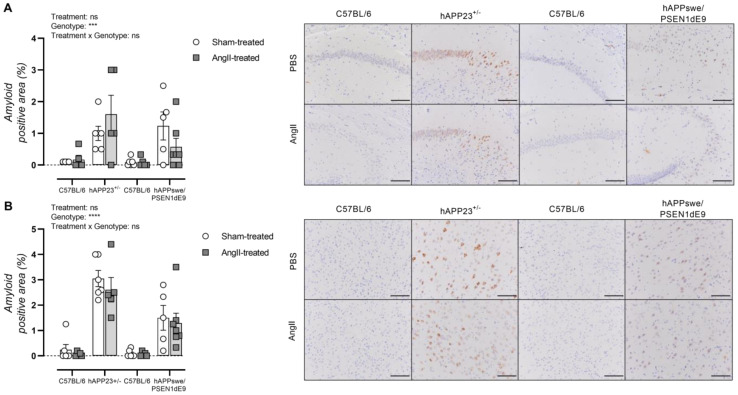
Histological assessment of cerebral amyloidosis in the (**A**) hippocampus and (**B**) cortex of sham- and AngII-treated C57BL/6 (sham: *n* = 6; AngII: *n* = 5) and hAPP223^+/−^ (sham: *n* = 6; AngII: *n* = 5) and C57BL/6 (sham: *n* = 6; AngII: *n* = 5) and hAPPswe/PSEN1dE9 (sham: *n* = 5; AngII: *n* = 6) brains. Factorial ANOVA for the factor’s ‘Treatment’, ‘Genotype’ and ‘Treatment × Genotype’ (*** *p* < 0.001, **** *p* < 0.0001). Data are presented as mean ± SEM. Scale bars indicate 100 µm.

**Table 1 ijms-23-02738-t001:** General information about animals. The *p*-value columns indicate the factorial effect of ‘Treatment’, ‘Genotype’ and ‘Treatment × Genotype’ as calculated with a factorial ANOVA (** *p* < 0.01, *** *p* < 0.001). Statistical differences between AngII-treated and sham-treated C57BL/6 animals are indicated by the ‘£’ symbol as the result of the respective Sidak post hoc test (^£^ *p* < 0.05). Statistical differences between AngII-treated and sham-treated AD murine models are indicated by the ‘$’ symbol as the result of the respective Sidak post hoc test (^$^ *p* < 0.05). Differences in AAA incidence between AngII-treated and sham-treated groups were calculated with a binomial test of the AngII-treated group against the respective sham-treated group with a 50/50% expectation (*** *p* < 0.0001). Non-significant results are indicated as ‘ns’. Data are presented as mean ± SEM. AAA = Abdominal aortic aneurysm.

	C57BL/6		hAPP23^+/−^				
	PBS (*n* = 14)	AngII (*n* = 12)	PBS (*n* = 11)	AngII (*n* = 11)	*p* _treatment_	*p* _genotype_	*p* _treatment × genotype_
Body weight (g)	27 ± 1	27 ± 1	27 ± 1	24 ± 1 ^$^	ns	**	ns
Survival, genotype (%)	74	-	80	-	-	ns	-
Survival, treatment (%)	100	57	76	80	-	ns	-
AAA incidence (%)	-	25	-	31	ns	-	-
	**C57BL/6**		**hAPPswe/** **PSEN1dE9**				
	**PBS** **(*n* = 14)**	**AngII** **(*n* = 13)**	**PBS** **(*n* = 12)**	**AngII** **(*n* = 11)**	** *p* _treatment_ **	** *p* _genotype_ **	** *p* _treatment × genotype_ **
Body weight (g)	34 ± 1	32 ± 1 ^£^	33 ± 1	32 ± 1	**	ns	ns
Survival, genotype (%)	90	-	75	-	-	ns	-
Survival, treatment (%)	100	58	95	72	-	0.09	-
AAA incidence (%)	-	69	-	31	***	-	-

**Table 2 ijms-23-02738-t002:** Systolic and diastolic heart function analysis of sham- and AngII-treated C57BL/6 and hAPP23^+/−^ mice. The *p*-value columns indicate the factorial effect of ‘Treatment’,‘Genotype’ and ‘Treatment × Genotype’ as calculated with a factorial ANOVA (* *p* < 0.05, ** *p* < 0.01, *** *p* < 0.001). Statistical differences between AngII-treated and sham-treated C57BL/6 animals are indicated with the ‘£’ symbol as the result of the respective Sidak post hoc test (^£^ *p* < 0.05). Statistical differences between AngII-treated and sham-treated AD murine models are indicated by the ‘$’ symbol as the result of the respective Sidak post hoc test (^$$$^ *p* < 0.001). Non-significant results are indicated as ‘ns’. Data presented as mean ± SEM. IVS,d = inner-ventricular septum thickness during diastole, LVID,d = left ventricular inner diameter during diastole, LVPW,d = left ventricular posterior wall thickness during diastole, LV mass/BW = left ventricular mass corrected for body weight, EF = ejection fraction, FS = fractional shortening, E/A = peak velocity blood flow in early diastole (E) to peak velocity flow in late diastole (A), E/E’ = mitral peak velocity of early filling (E) to early diastolic mitral annular velocity (E’), IVRT = isovolumetric retention time.

	C57BL/6		hAPP23^+/−^				
	PBS (*n* = 14)	AngII (*n* = 12)	PBS (*n* = 11)	AngII (*n* = 11)	*p* _treatment_	*p* _genotype_	*p* _treatment × genotype_
Heart weight (mg)	165 ± 5	188 ± 5 ^£^	141 ± 3	183 ± 8 ^$$$^	***	*	ns
Heart weight (%)	0.53 ± 0.01	0.58 ± 0.01	0.51 ± 0.01	0.69 ± 0.04 ^$$$^	***	*	**
IVS,d (mm)	1.2 ± 0.1	1.1 ± 0.1	1.1 ± 0.1	1.3 ± 0.1	ns	ns	ns
LVID,d (mm)	3.7 ± 0.1	3.7 ± 0.1	3.6 ± 0.1	3.4 ± 0.2	ns	0.06	ns
LVPW,d (mm)	1.1 ± 0.1	1.1 ± 0.1	1.1 ± 0.1	1.4 ± 0.1	0.08	0.07	ns
LV mass (mg)	172 ± 11	167 ± 10	157 ± 11	187 ± 18	ns	ns	ns
LV mass/BW (10^−3^)	5.5 ± 0.3	5.2 ± 0.3	5.7 ± 0.5	7.1 ± 0.7	ns	*	ns
LV volume,d (µL)	60 ± 4	61 ± 5	54 ± 4	49 ± 6	ns	0.08	ns
Stroke volume (µL)	42 ± 3	42 ± 3	39 ± 2	33 ± 3	ns	0.08	ns
EF (%)	71 ± 2	70 ± 3	73 ± 2	71 ± 3	ns	ns	ns
FS (%)	40 ± 2	39 ± 2	42 ± 2	40 ± 2	ns	ns	ns
E/A (none)	1.4 ± 0.1	1.4 ± 0.1	1.5 ± 0.3	1.6 ± 0.1	ns	ns	ns
E/E’ (none)	32 ± 3	29 ± 4	35 ± 4	51 ± 7	ns	*	ns
IVRT (ms)	19 ± 1	21 ± 1	23 ± 2	24 ± 3	ns	0.07	ns
Deceleration (ms)	17 ± 1	15 ± 1	16 ± 2	21 ± 2	ns	ns	ns

**Table 3 ijms-23-02738-t003:** Systolic and diastolic heart function analysis of sham- and AngII-treated C57BL/6 and hAPPswe/PSEN1dE9 mice. The *p*-value columns indicate the factorial effect of ‘Treatment’, ‘Genotype’ and ‘Treatment × Genotypee’ as calculated with a factorial ANOVA (* *p* < 0.05, ** *p* < 0.01, *** *p* < 0.001). Statistical differences between AngII-treated and sham-treated C57BL/6 animals are indicated with the ‘£’ symbol as the result of the respective Sidak post hoc test (^£^ *p* < 0.05, ^£££^ *p* < 0.001). Statistical differences between AngII-treated and sham-treated AD murine models are indicated by the ‘$’ symbol as the result of the respective Sidak post hoc test (^$$$^ *p* < 0.001). Non-significant results are indicated as ‘ns’. Data presented as mean ± SEM. IVS,d = inner-ventricular septum thickness during diastole, LVID,d = left ventricular inner diameter during diastole, LVPW,d = left ventricular posterior wall thickness during diastole, LV mass/BW = left ventricular mass corrected for body weight, EF = ejection fraction, FS = fractional shortening, E/A = peak velocity blood flow in early diastole (E) to peak velocity flow in late diastole (A), E/E’ = mitral peak velocity of early filling (E) to early diastolic mitral annular velocity (E’), IVRT = isovolumetric retention time.

	C57BL/6		hAPPswe/ PSEN1dE9				
	PBS (*n* = 14)	AngII (*n* = 13)	PBS (*n* = 12)	AngII (*n* = 11)	*p* _treatment_	*p* _genotype_	*p* _treatment × genotype_
Heart weight (mg)	182 ± 5	218 ± 7 ^£££^	158 ± 3	204 ± 9 ^$$$^	***	**	ns
Heart weight (%)	0.53 ± 0.01	0.68 ± 0.02 ^£££^	0.48 ± 0.01	0.65 ± 0.02 ^$$$^	***	*	ns
IVS,d (mm)	1.2 ± 0.1	1.4 ± 0.1	1.1 ± 0.1	1.3 ± 0.1	*	ns	ns
LVID,d (mm)	3.6 ± 0.1	3.8 ± 0.1	3.6 ± 0.1	3.6 ± 0.2	ns	ns	ns
LVPW,d (mm)	1.0 ± 0.1	1.2 ± 0.1	1.0 ± 0.1	1.1 ± 0.1	*	ns	ns
LV mass (mg)	170 ± 21	224 ± 17	148 ± 15	186 ± 18	*	ns	ns
LV mass/BW (10^−3^)	4.9 ± 0.6	7.0 ± 0.5 ^£^	4.5 ± 0.5	5.9 ± 0.5	**	ns	ns
LV volume,d (µL)	57 ± 4	63 ± 4	55 ± 4	54 ± 6	ns	ns	ns
Stroke volume (µL)	42 ± 3	42 ± 2	38 ± 3	39 ± 3	ns	ns	ns
EF (%)	73 ± 2	68 ± 3	70 ± 2	74 ± 4	ns	ns	ns
FS (%)	42 ± 2	38 ± 3	39 ± 2	43 ± 4	ns	ns	ns
E/A (none)	1.6 ± 0.2	1.7 ± 0.1	1.4 ± 0.3	1.5 ± 0.1	ns	ns	ns
E/E’ (none)	32 ± 5	23 ± 4	31 ± 5	22 ± 4	ns	ns	ns
IVRT (ms)	21 ± 2	22 ± 1	22 ± 1	24 ± 3	ns	ns	ns
Deceleration (ms)	16 ± 1	18 ± 2	17 ± 1	17 ± 1	ns	ns	ns

## Data Availability

Not applicable.

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
