# Peer review of "Short-Term Pharmacological Induction of Arterial Stiffness and Hypertension with Angiotensin II Does Not Affect Learning and Memory and Cerebral Amyloid Load in Two Murine Models of Alzheimer’s Disease"

_ijms, 2022, doi:10.3390/ijms23052738_

Round 1

Reviewer 1 Report

Hendrickx and colleagues in the present study entitled ‘Short-term pharmacological induction of arterial stiffness and hypertension with angiotensin II does not affect learning and memory and cerebral amyloid load in two murine models of Alzheimer’s disease’, investigated the current status of knowledge of short-term angiotensin II treatment to induce hypertension and the possible effect on learning and memory and amyloid load in Alzheimer’s disease using two mouse models (hAPP23+/- and 18 hAPPswe/PSEN1dE9 mice). The results showed increased arterial stiffness in vivo and hypertension pharmacological treatment; however, visuospatial learning and memory and pathological amyloid load in both Alzheimer’s disease mouse models were not further impaired. The authors concluded by stating that probably the treatment period with angiotensin II was too short to observe additional effects on cognition.

The main strength of this original research article is that it addresses an interesting and timely question, investigating the linkage between arterial stiffness and/or hypertension with Alzheimer’s disease. In general, I think the idea of this article is really interesting and the authors’ fascinating observations on this timely topic may be of interest to the readers of the International Journal of Molecular Sciences. However, some comments, as well as some crucial evidence that should be included to support the authors’ argumentation, needed to be addressed to improve the quality of the article, its adequacy, its replicability, and thus its readability prior to the publication in the present form. My overall judgment is to publish this article after the authors have carefully considered my suggestions below, in particular reshaping parts of the Introduction and Discussion sections by adding more evidence. Moreover, I believe that some adjustments are necessary in the Materials and Methods section to provide in-depth understanding and replicability of the findings.

Please consider the following comments:

  • Abstract: I think that the lack of an explanation of arterial stiffness and the effects of its pharmacological induction on the progression of Alzheimer’s disease makes the readers unable to grasp the key aspects of this paper by consulting the abstract.
  • Keywords: Please consider adding ‘hypertension’ and ‘angiotensin II’ as keywords.
  • In general, I recommend authors to use more evidence to back their claims, especially in the Introduction of the article, which I believe is currently lacking. Thus, I recommend the authors to attempt to deepen the subject of their manuscript, as the bibliography is too concise: nonetheless, in my opinion, less than 50 articles for a research paper are really insufficient. Indeed, currently, authors cite only 44 papers, and they are too low. Therefore, I suggest the authors to focus their efforts on researching more relevant literature: I believe that adding more studies and reviews will help them to provide better and more accurate background to this study. In this review, I will try to help the authors by suggesting some relevant literature of my knowledge that suits their manuscript.
  • Page 1, Introduction: In this section, the authors take a rather narrow view of age-related disorders, and emphasized the connection between cardiovascular disease (CVD) and dementia. Nevertheless, I think that what is really missing is a scientific overview on Alzheimer’s disease (AD), that helps readers increasing their knowledge about AD definition, causes, symptoms and related neurocognitive changes. Considering that this study's main focus is to deepen current understanding of specific hallmarks in the prodromal stage of the AD disease, I suggest the authors to make such effort to provide a brief overview of the pertinent published literature that offer a perspective on pathophysiology and neurological changes of AD, because as it stands, there is no mention of this in the manuscript. In this regard, I believe that the statement ‘…dementia was the fifth leading cause of death worldwide with Alzheimer’s disease (AD) being the most prominent dementia syndrome’ needs some necessary citations. In particular, according with this sentence, I would suggest some crucial references that will methodologically fit with the present manuscript, for example, a recent review that examined pathophysiological basis and biomarkers of AD pathology and investigated molecular signs of neuroinflammation in neurodegenerative diseases, in particular Alzheimer’s disease (https://doi.org/10.3390/ijms21072431). Importantly, I also recommend a relevant study in which authors investigated age-related impairments in the ability to process contextual information and in the regulation of responses to threat, addressing that structural and physiological alterations in the prefrontal cortex and medial temporal lobe determine cognitive changes in advanced aging, that can eventually cause patterns of cognitive dysfunctions observed in patients with AD/MCI (https://doi.org/10.1038/s41598-018-31000-9). I firmly believe that these references will help to provide a more coherent and defined background.
  • Page 1, Introduction: In according with the previous point raised, when authors stated that ‘clinical research has associated hypertension with increased cerebral amyloid-β (Aβ) plaque burden, neurofibrillary tangle density and cerebral atrophy’, I would suggest adding some studies that might address how hypertension increases amyloid-β (Aβ) pathology, highlighting the combined effect of forms of Aβ and tau protein to drive healthy neurons into the diseased state. Aβ peptide and tau protein consistently accumulate in the frontal and/or parietal lobes, and cause alterations of the frontal lobe that impact memory and error-driven learning in individuals who have a high risk of dementia: evidence from a recent study conducted on patients with a lesion in ventromedial portion of the prefrontal cortex (https://doi.org/10.1523/JNEUROSCI.0304-20.2020) revealed that the ventromedial prefrontal cortex (vmPFC) is involved in the acquisition of emotional conditioning (i.e., learning), assessing how naturally occurring bilateral lesion centered on the vmPFC compromises the generation of a conditioned psychophysiological response during the acquisition of pavlovian threat conditioning (i.e., emotional learning). Also, in a recent theoretical review (https://doi.org/10.1038/s41380-021-01326-4) that focused on the neurobiology of memory emotional conditioning, the role of the ventromedial prefrontal cortex (vmPFC) was analyzed in the processing of safety-threat information and their relative value, and how this region is fundamental for the evaluation and representation of stimulus-outcome’s value needed to produce sustained physiological responses. Secondary, authors also might to consider some studies that have focused on this topic. I believe that adding information from these studies may improve the theoretical background of the present article and its argumentation by highlighting how cognitive alterations caused by frontal dysfunction are fundamental as neurodegenerative biomarkers of AD.
  • The objectives are generally clear; however, I believe that there are some ambiguous points that require clarification or refining. I think that authors here need to be explicit regarding how they operationally determined the effects that pharmacological induction of AS and hypertension had on visuospatial learning and memory abilities. Furthermore, I think that the authors should be explicit regarding the hypothesis of a direct link between hypertension and amyloidosis.
  • Page 2, Results: In my opinion, this section is well organized, but it illustrates findings in an excessively broad way, without really providing full statistical details, to ensure in-depth understanding and replicability of the findings. Indeed, in my opinion, it is necessary for the authors to present their findings with a precise description not just in tables, but also in the main text.
  • Page 9, Discussion: In this final section, the authors described the results and their argumentation and captured the state of the art well; however, I would have liked to see some views on a way forward. I believe that the authors should make an effort, trying to explain the theoretical implication as well as the translational application of this research article, to adequately convey what they believe is the take-home message of their study. Discussion of theoretical and methodological avenues in need of refinement is necessary, as well as suggestions of a path forward in understanding the connection between hypertension onset and cognitive decline in AD, and new possibilities of treatment for patients at risk of AD. In this regard, recent evidence suggests that the application of new methods in Alzheimer’s treatment, such as the Non-invasive brain stimulation techniques (NIBS), have shown promising results in humans. Importantly, I recommend referring to recent studies that revealed that the application of NIBS induces long-lasting effects, noninvasively modulating the cortical excitability, and modulating a variety of cognitive functions: for example, a recent review acknowledged the implementation of NIBS to modulate in general fear memories (https://doi.org/10.1016/j.neubiorev.2021.04.036). Additionally, I would suggest another recent review that illustrated the therapeutic potential of NIBS as a valid alternative for those patients not responding or drug treatments, in this review authors discussed to use ‘State-dependent Transcranial Magnetic Stimulation for the treatment of specific phobia’. In addition to the previously mentioned literature, authors might also see these additional studies that have focused on the efficacy of NIBS and IBS (https://doi.org/10.3389/fpsyt.2018.00201; https://doi.org/10.3389/fnagi.2020.578339).
  • In my opinion, I think the ‘Conclusions’ paragraph would benefit from some thoughtful as well as in-depth considerations by the authors, because as it stands, it is very descriptive but not enough theoretical as a discussion should be. Authors should make an effort, trying to explain the theoretical implication as well as the translational application of their research.
  • In according to the previous comment, I would ask the authors to include a ‘Limitations and future directions’ section before the end of the manuscript, in which authors can describe in detail and report all the technical issues brought to the surface.
  • Figures: I suggest to modify all figures for clarity and provide higher-quality images because, as it stands, the readers may have difficulty comprehending them. In my opinion, data settings are written with a very small font. Also, please change the scale of the vertical axis and use the same minimum/maximum scale value in all the graphs in all the figures and reorganize the graphs’ space, to provide a better understanding and a direct interpretation of the results. Finally, I suggest adding a figure that can clearly describes experimental design, specifically showing the visuospatial learning and memory test used in mice.
  • References: According to the Journal’s guidelines, authors should have provided the DOI number for each reference.

Overall, the manuscript contains 5 figures, 3 tables, and 44 references. In my opinion, the number of references (and thus evidence) are exiguous for an original research article, and this issue may prevent the possibility of publishing it in this form. However, I believe that the manuscript may carry important value in investigating the linkage between arterial stiffness and/or hypertension with Alzheimer’s disease.

I hope that, after these careful revisions, the manuscript can meet the Journal’s high standards for publication. I am available for a new round of revision of this review.

Best regards,

Reviewer

Reviewer 2 Report

The work by  Hendrickx et al. is focused on the overlapping between cardiovascular diseases and Alzheimer’s disease. In particular Authors investigate the correlation between pharmacological induced (by Angiotensin II) arterial stiffness and Alzheimer’s disease. Authors showed that short-term treatment with angiotensin II, although induces hypertension does not affect amyloid load and/or learning and memory effect on two AD murine models. Overall, experiments were well designed and conclusions are well supported by results. For these reasons I suggest the paper for publication in IJMS.

Author Response

The authors would like to thank Reviewer 2 for the time and effort he/she invested in reviewing the manuscript and the resulting positive feedback. 

Round 2

Reviewer 1 Report

I am very pleased to see that the authors have welcomed my suggestions and have clarified several of the questions I raised in my first round of this review. I believe that this work does an excellent job demonstrating the linkage between arterial stiffness and/or hypertension with Alzheimer’s disease, and presenting interesting data showing that increased arterial stiffness in vivo and hypertension after pharmacological treatment with angiotensin II, but no further impairment in cognition in both Alzheimer’s disease mouse models used.

I only have one last minor suggestion to do, to further improve the theoretical background of the present article and its argumentation by highlighting how cognitive alterations caused by frontal dysfunction are fundamental as neurodegenerative biomarkers of AD. I suggest including a very recent perspective manuscript (https://doi.org/10.17219/acem/146756) that has focused on providing a deeper understanding of human learning neural networks, showed the crucial role of human PFC, giving interesting insights on the involvement of this important brain region in the advancement of alternative, more precise and individualized treatments for a variety of neurologic and psychiatric disorders.

Overall, this is a timely and needed study, and I look forward to seeing further study on this issue by these authors in the future.

Author Response

We would like to thank Reviewer 1 for the suggestion and the provision of this recent research. We adapted our text accordingly on lines 48 - 56 of the manuscript.